System wide analyses have underestimated protein abundances and the importance of transcription in mammals

Li Jingyi Jessica 1 2
Bickel Peter J 1
Biggin Mark D 3 mdbiggin@lbl.gov
1 Department of Statistics, University of California , Berkeley, CA , USA
2 Departments of Statistics and Human Genetics, University of California , Los Angeles, CA , USA
3 Genomics Division, Lawrence Berkeley National Laboratory , Berkeley, CA , USA
Engelhardt Barbara
Electronic publication date: 2014 Feb 27
Publication date: 2014
Volume: 2
Electronic Location ID: e270
Received 2013 Sep 11; Accepted 2014 Jan 22
License: This is an open access article, free of all copyright, made available under the Creative Commons Public Domain Dedication. This work may be freely reproduced, distributed, transmitted, modified, built upon, or otherwise used by anyone for any lawful purpose.
License URL: https://creativecommons.org/publicdomain/zero/1.0/

Keywords: Transcription, Translation, Mass spectrometry, Gene expression, Protein abundance

Funding: This work was supported in part by NIH grant P01 GM009655. Work at Lawrence Berkeley National Laboratory was conducted under Department of Energy contract DEAC02-05CH11231. The funders had no role in study design, data collection and analysis, decision to publish, or preparation of the manuscript.

==============================
Large scale surveys in mammalian tissue culture cells suggest that the protein expressed at the median abundance is present at 8,000–16,000 molecules per cell and that differences in mRNA expression between genes explain only 10–40% of the differences in protein levels. We find, however, that these surveys have significantly underestimated protein abundances and the relative importance of transcription. Using individual measurements for 61 housekeeping proteins to rescale whole proteome data from Schwanhausser et al. (2011), we find that the median protein detected is expressed at 170,000 molecules per cell and that our corrected protein abundance estimates show a higher correlation with mRNA abundances than do the uncorrected protein data. In addition, we estimated the impact of further errors in mRNA and protein abundances using direct experimental measurements of these errors. The resulting analysis suggests that mRNA levels explain at least 56% of the differences in protein abundance for the 4,212 genes detected by Schwanhausser et al. (2011), though because one major source of error could not be estimated the true percent contribution should be higher. We also employed a second, independent strategy to determine the contribution of mRNA levels to protein expression. We show that the variance in translation rates directly measured by ribosome profiling is only 12% of that inferred by Schwanhausser et al. (2011), and that the measured and inferred translation rates correlate poorly (R2 = 0.13). Based on this, our second strategy suggests that mRNA levels explain ∼81% of the variance in protein levels. We also determined the percent contributions of transcription, RNA degradation, translation and protein degradation to the variance in protein abundances using both of our strategies. While the magnitudes of the two estimates vary, they both suggest that transcription plays a more important role than the earlier studies implied and translation a much smaller role. Finally, the above estimates only apply to those genes whose mRNA and protein expression was detected. Based on a detailed analysis by Hebenstreit et al. (2012), we estimate that approximately 40% of genes in a given cell within a population express no mRNA. Since there can be no translation in the absence of mRNA, we argue that differences in translation rates can play no role in determining the expression levels for the ∼40% of genes that are non-expressed.

Introduction

The protein products of genes are expressed at very different levels from each other in a mammalian cell. Thousands of genes are not detectably expressed. Of those that are, their proteins are present at levels that differ by five orders of magnitude. Cytoplasmic actin, for example, is expressed at 1.5 × 108 molecules per cell (Kislauskis et al., 1997), whereas some transcription factors are expressed at only 4 × 103 molecules per cell (Biggin, 2011). There are four major steps that determine differences in protein expression: the rates at which genes are transcribed, mRNAs are degraded, proteins are translated, and proteins are degraded (Fig. 1). The combined effect of transcription and mRNA degradation together determines mRNA abundances (Fig. 1). The joint effect of protein translation and protein degradation controls the differences between mRNA and protein concentrations (Fig. 1).

Figure 1 The steps regulating protein expression.

The steady state abundances of mRNAs and proteins are each determined by their relative rates of production (i.e., transcription or translation) and their rates of degradation.

Transcription has long been regarded as a dominant step and is controlled by sequence specific transcription factors that differentially interact with cis-regulatory DNA regions. The rates of the other three steps, however, vary significantly between genes as well (Boisvert et al., 2012; Cambridge et al., 2011; Cheadle et al., 2005; de Sousa Abreu et al., 2009; Eden et al., 2011; Guo et al., 2010; Han et al., 2014; Hentze and Kuhn, 1996; Hsieh et al., 2012; Ingolia et al., 2011; Kristensen et al., 2013; Loriaux & Hoffmann, 2013; Rabani et al., 2011; Schwanhausser et al., 2011; Sharova et al., 2009; Yang et al., 2003). MicroRNAs, for example, differentially interact with mRNAs to alter rates of RNA degradation and protein translation (Ambros, 2011; Baek et al., 2008; Elmen et al., 2008; Gennarino et al., 2012; Guo et al., 2010; Hobert, 2008; Krutzfeldt et al., 2005; Pillai et al., 2007; Rajewsky, 2011; Selbach et al., 2008; Subtelny et al., in press; Xiao et al., 2007).

To quantify the relative importance of each of the four steps, label free mass spectrometry methods have been developed that measure the absolute number of protein molecules expressed per cell for thousands of genes (Bantscheff et al., 2012; Beck et al., 2011; Maier, Guell & Serrano, 2009; Schwanhausser et al., 2011; Vogel et al., 2010; Vogel & Marcotte, 2012). By comparing these data to mRNA abundance data, one can determine the importance of transcription and mRNA degradation combined versus the importance of protein translation and protein degradation combined (Maier, Guell & Serrano, 2009; Schwanhausser et al., 2011; Vogel & Marcotte, 2012) (Fig. 1). By measuring mRNA degradation and protein degradation rates as well, the rates of transcription and translation can be additionally inferred indirectly. Using this approach to study mouse NIH3T3 fibroblasts, Schwanhausser et al. (2011) concluded that mRNA levels explain ∼40% of the variability in protein levels; that the cellular abundance of proteins is predominantly controlled at the level of translation; that transcription is the second largest determinant; and that the degradation of mRNAs and proteins play a significant but lesser role.

The above work has provided critically important datasets and an initial framework for analysis. We noticed, however, that Schwanhausser et al.’s (2011) protein abundance estimates are mostly lower than established values for individual proteins in the literature and that statistical methods to quantitate the impact of experimental error had not been employed. We therefore set out to explore if we could refine the analysis of these datasets and to compare our results to those of Schwanhausser et al. (2011) and other systemwide studies.

Results and discussion

A non-linear underestimation of protein abundances

Our starting point was a set of published abundances of 53 mammalian housekeeping proteins, most of which are based on SILAC mass spectrometry or western blot data (Biggin, 2011; Brosi, Hauri & Kramer, 1993; Gregory et al., 2002; Hanamura et al., 1998; Kimura et al., 1999; Kislauskis et al., 1997; Princiotta et al., 2003; Wollfe, 1998; Wong et al., 2011; Zeiler et al., 2012). On average these established estimates are 16 fold higher than those from Schwanhausser et al.’s (2011) original label free mass spectrometry data (Dataset S1). Once we brought this discrepancy to the authors’ attention, they upwardly revised their label free abundance estimates for all 5,028 detected proteins and in addition provided western blot or Selected Reaction Monitoring (SRM) mass spectrometry measurements for eight polypeptides in NIH3T3 cells (see Corrigendum; Schwanhausser et al., 2011). However, Schwanhausser et al.’s (2011) second whole proteome abundance estimates are still lower than individual measurements for proteins expressed below 106 molecules per cell, with the lowest abundance proteins showing the largest discrepancy (Fig. 2A; Dataset S1).

Figure 2 A non-linear bias in protein abundance estimates and its correction.

(A) The y axis shows the ratios of 61 individually derived protein abundance estimates each divided by the corresponding abundance estimate from Schwanhausser et al.’s (2011) second whole proteome dataset. The x axis shows the abundance estimate from Schwanhausser et al.’s (2011) second whole proteome dataset. The red line indicates the locally weighted line of best fit (lowess parameter f = 1.0), and the vertical dotted grey lines show the locations of the 1st quartile, median and 3rd quartile of the abundance distribution of the 5,028 proteins detected in the whole proteome analysis. (B) The same as panel A. except that the whole proteome estimates of Schwanhausser et al. (2011) have been corrected using a two-part linear model and the abundances from the 61 individual protein measurements, see Fig. 3B.

Western blot and SILAC mass spectrometry measurements show the same discrepancy versus the label free whole proteome data (Dataset S1). For example, for proteins expressed below 1 million molecules per cell, the 26 SILAC measurements are a median of 2.95 fold higher than Schwanhausser et al.’s (2011) second estimates, and the 19 western blot measurements are 3.10 fold higher. This suggests that the discrepancy is not due to error in the individual measurements as a similar bias in two independent methods is unlikely.

Of the 61 individual measurements of protein abundance available to us, 15 were made in NIH3T3 cells and 42 were made in HeLa cells. The discrepancy between Schwanhausser et al.’s (2011) second whole proteome abundances and these individual measurements is not due to differences in expression levels between HeLa and NIH3T3 cells for the following reasons. One, it is unlikely that such a difference would only occur for lower abundance proteins. Two, five of the individual measurements for lower abundance proteins (Orc2, Orc4, HDAC3, NFkB1, and NFkB2) were made in NIH3T3 cells and are on average 3.7 fold higher than the second whole proteome estimates in this same cell line (Dataset S1). Three, later in the paper we show that collectively the 61 individual proteins measured have on average the same relationship in expression values versus all other cellular proteins in both NIH3T3 and HeLa cells. Finally, Schwanhausser et al.’s (2011) second estimates for RNA polymerase II and general transcription factors such as TFIIB and TFIIE are only 1.6 fold higher than those in yeast (Borggrefe et al., 2001) and are 7.1 times less than those in HeLa cells (Kimura et al., 1999). Yeast cells have 1/40th the volume, 1/200th the amount of DNA and 1/4 the number of genes of NIH3T3 and HeLa cells (Milo et al., 2010). Two fold reductions in the concentrations of a single general transcription factor have, in some cases, phenotypic consequence (Aoyagi & Wassarman, 2001; Deutschbauer et al., 2005; Eissenberg et al., 2002; Kim et al., 2010). Thus, it is unlikely that a rapidly dividing mammalian cell could function with much larger reductions in the amounts of all of these essential regulators to levels close to those found in yeast.

Correcting the non-linear bias

Schwanhausser et al. (2011) calibrated protein abundances by spiking known amounts of protein standards into a crude protein extract from NIH3T3 cells and then measuring the abundances of several thousand proteins in the mixture by iBAC label free mass spectrometry. The 20 ‘spiked in’ protein standards detected in this experiment, however, were present at the equivalent >8.0 × 105 molecules per cell, a level that represents only the most highly expressed 11% of the proteins detected (Fig. 3A) (M Selbach, personal communication; Schwanhausser et al., 2011). To convert mass spectrometry signals to protein abundances, Schwanhausser et al. (2011) assumed that a linear relationship defined using the 20 ‘spiked in’ standards holds true for proteins at all abundances (Fig. 3A). The discrepancy between the resulting estimates and individual protein measurements (Fig. 2A), however, suggests that this assumption is not valid. A recent benchmarking study also supports this conclusion, showing that in general in the iBAC method ‘low-abundance proteins were dramatically underestimated’ (Ahrne et al., 2013). We therefore employed the 61 individual protein measurements from the literature as they span a much wider abundance range. In a plot of these data versus Schwanhausser et al.’s (2011) second whole proteome estimates, we found that a two-part linear regression gave a statistically better fit over a single regression (Figs. 3B and 3C) (p-value = 0.002, Materials and Methods). We then used this two-part regression to derive new abundance estimates for all 5,028 proteins in Schwanhausser et al.’s (2011) dataset (Dataset S1). As Fig. 2B shows, the correction removes the non-linear bias.

Figure 3 Calibrating absolute protein abundances.

(A) The relationship between iBAC mass spectrometry signal (x axis) and the amounts of the 20 ‘spiked in’ protein standards (y axis) used by Schwanhausser et al. (2011) to calibrate their whole proteome abundances (data kindly provided by Matthias Selbach, Dataset S2). The line of best fit is shown (red). (B) The relationship between individually derived estimates for 61 housekeeping proteins (y axis) and Schwanhausser et al.’s (2011) second whole proteome estimates (x axis). The two part line of best fit used to correct the second whole proteome estimates is shown (solid red line) as is the single linear regression (dashed red line). (C) The fit of different regression models for the data in panel b. The y axis shows the leave-one-out cross validation root mean square error for each model. The x axis shows the protein abundance used to separate the data for two part linear regressions. The red curve shows the optimum change point for a two part linear model is at an abundance of ∼106 molecules per cell. The dashed red horizontal line shows the root mean square error for the single linear regression.

In our rescaled data, the median abundance protein is present at 170,000 molecules per cell (Fig. 2B), considerably higher than Schwanhausser et al.’s (2011) original estimate of 16,000 molecules per cell and significantly above their second estimate of 50,000 molecules per cell. For low abundance proteins the effect is larger. In our corrected data, the median sequence specific transcription factor is present at 71,000 molecules per cell versus Schwanhausser et al.’s (2011) estimates of first 3,500 then 9,300 molecules per cell (Dataset S1). Our correction reduces the range of detected abundances by ∼50 fold (unlogged) compared to Schwanhausser et al.’s second estimates (Dataset S1) and the variance in protein levels from 0.97 (log10) to 0.36 (log10).

Corrected protein abundances show an increased correlation with mRNA abundances

As an independent check on the accuracy of our corrected abundances, we compared them to Schwanhausser et al.’s (2011) RNA-Seq mRNA expression data. Our corrected protein abundances correlate more highly with mRNA abundances than do Schwanhausser et al.’s (2011) second whole proteome estimates (compare Figs. 4A and 4B). The increase in correlation coefficient is highly significant (p-value < 10-29) (Materials and Methods), arguing that our non-linear correction to the whole proteome abundances has increased the accuracy of these estimates. The most dramatic change is that the scatter about the line of best fit is reduced and shows a stronger linear relationship. The 50% prediction band shows that prior to correction the half of proteins whose abundances are best predicted by mRNA levels are expressed over an 11 fold range (unlogged), but after correction they are expressed over a narrower, 4 fold range (Figs. 4A and 4B). The correction reduces the width of the 95% prediction band even further, by 18 fold.

Figure 4 Protein abundance estimates versus mRNA abundances.

(A) The relationship between Schwanhausser et al.’s (2011) second protein abundance estimates versus mRNA levels for 4,212 genes in NIH3T3 cells. The linear regression of the data is shown in red, the 50% prediction band by dashed green lines, and the 95% prediction band by dashed blue lines. (B) The relationship between our corrected estimates of protein abundance versus mRNA levels. The linear regression and prediction bands are labeled as in panel A.

For our corrected data, the median number of proteins translated per mRNA is 9,800 compared to Schwanhausser et al.’s (2011) original estimate of 900 and their second estimate of 2,800. In yeast, the ratio of protein molecules translated per mRNA is 4,200–5,600 (Ghaemmaghami et al., 2003; Lu et al., 2007). Given that mammalian cells have a higher protein copy number than yeast (Milo et al., 2010), it is not unreasonable that the ratio in mammalian cells would be higher.

Estimating the impact of molecule specific measurement error

In addition to the above general error in scaling protein abundances, there are additional sources of experimental error that uniquely affect data for each protein and mRNA differently. As a result of these molecule specific measurement errors, the coefficient of determination between measured mRNA and measured protein levels—i.e., R2 shown in Fig. 4B—is lower than the actual value between true protein and true mRNA levels. With an accurate estimate of the errors, it is possible to calculate the increased correlation expected between true protein and true mRNA abundances. Because the variance in the residuals in Fig. 4B (i.e., the displacement along the y axis of data points about the line of best fit) is composed of both experimental error and the genuine differences in the rates of translation and protein degradation between genes, once the experimental error has been estimated, it is also possible to infer the combined true effects of translation and protein degradation.

There are two classes of molecule specific experimental error: stochastic and systematic. Stochastic error, or imprecision, is the variation between replica experiments and is estimated from this variation. Systematic error, or inaccuracy, is the reproducible under or over estimation of each data point, and is estimated by comparing the results obtained with the assay being used to those from gold standard measurements obtained with the most accurate method available.

Schwanhausser et al. (2011) limited their estimation of experimental error to stochastic errors. Because our correction of the whole proteome abundances reduces the total variance in measured protein expression levels, we first reestimated the proportion of the variance in the residuals in Fig. 4B that is due to stochastic measurement error using replica datasets (Materials and Methods). We find that 7% of this variance results from stochastic protein error and 0.8% from stochastic mRNA error.

Schwanhausser et al. (2011), however, also noted a significant variance between their whole genome RNA-Seq data and NanoString measurements for 79 genes (R2 = 0.79 in Figure S8(A) in Schwanhausser et al., 2011), though they did not take this into account subsequently. RNA-Seq is well known to suffer reproducible several fold biases in the number of DNA sequence reads obtained for different GC content genomic regions (Cheung et al., 2011; Dohm et al., 2008). In contrast, NanoString gives an accurate measure of nucleic acid abundance as correlation coefficients of R2 = 0.99 are obtained when NanoString data are compared to known concentrations of nucleic acid standards (Geiss et al., 2008). Thus, it is reasonable to consider NanoString as a gold standard that can be used to assess the systematic error in the RNA-seq data by assuming that the variance between the two methods is due mostly to systematic error in RNA-seq. Using Analysis of Variance (ANOVA), the variance in Schwanhausser et al.’s (2011) NanoString/RNA-Seq comparison can be shown to be equivalent to 23.3% of the variation in the residuals in Fig. 4B, 29 fold larger than the stochastic component of mRNA error (see Materials and Methods for a discussion of the assumptions used in this analysis).

It is also important to assess the systematic error in the whole proteome abundances as label free mass spectrometry includes such biases (Ahrne et al., 2013; Bantscheff et al., 2012; Kuntumalla et al., 2009; Lu et al., 2007; Peng et al., 2012). In principle the ‘spiked in’ protein standards in Schwanhausser et al.’s (2011) calibration experiment (i.e., the data in Fig. 3A) should provide gold standard data. In practice, however, the variance in mass spectrometry estimates for protein standards present at supposedly the same amounts is too high (i.e., the scatter along the x axis in Fig. 3A). This variance would contribute 61% to the variance in the residuals in Fig. 4B, yet the variance of the residuals between the corrected whole proteome estimates and the 61 individual protein measurements (i.e., the scatter along the x axis about the solid red line in Fig. 3B) would contribute only 44%. Since the western blot and SILAC methods used to make the 61 individual protein measurements introduce some experimental error, it seems likely that the commercial protein standards used by Schwanhausser et al. (2011) were not as accurately prepared at the correct protein concentrations as one would expect. Since no other suitable gold standard is available, we are thus unable to estimate the systematic protein error, though it is likely to be less than 44% of variance in the residuals in Fig. 4B.

Taking the stochastic protein error as a minimum estimate of protein error and the variance from the NanoString/RNA-Seq comparison as an estimate of all RNA errors, it can be shown that true mRNA levels explain at least 56% of true protein levels, and by extension protein degradation and translation combined explain no more than 44% (see Materials and Methods).

Estimating the relative importance of transcription, mRNA degradation, translation and protein degradation

In addition to determining protein and mRNA abundances, Schwanhausser et al. (2011) also directly measured mRNA and protein degradation rates and calculated the percentage that each contributed to the variance in protein abundances. Using this information, it is possible to determine the relative importance of transcription, RNA degradation, translation and protein degradation for different scenarios (Table 1, see Materials and Methods). For the 4,212 genes whose protein and mRNA expression was detected, our analysis suggests that transcription explains ∼38% of the variance in true protein levels, RNA degradation explains ∼18%, translation ∼30%, and protein degradation ∼14% (Table 1). Clearly these estimates are tentative and depend on the particular assumptions we have made. We believe, though, that they will prove more accurate than Schwanhausser et al.’s (2011) suggestion that translation is the predominant determinant of protein expression and that mRNA levels explain around 40% of the variability in protein levels (Table 1).

Table 1 The contribution of different steps in gene expression to the variance in protein abundances between genes.

	Variance in protein levels (log10)*	Percent contribution to variance in protein levels	
		mRNA(%)	Transcription(%)	RNA degradation(%)	Translation(%)	Protein degradation(%)	
Schwanhausser 2nd dataa	0.97	40	34	6	55	5	
Measured protein error strategyb	0.34	56	38	18	30	14	
Measured translation strategyc	0.61	81	71	10	11	8	
* In this column, the value given for Schwanhausser et al.’s (2011) 2nd data is the variance in their measured protein abundances; the remaining values are our estimate for the variance in true protein levels for different scenarios.

a Estimates from Schwanhausser et al. (2011) based on the 4,212 genes for which NIH3T3 cell protein and mRNA abundance data are available.

b Our estimates for the same 4,212 genes studied by Schwanhausser et al. (2011) after correcting the overall scaling of the NIH3T3 cell protein abundance data and taking several sources of molecule specific experimental error into account: stochastic protein error and all mRNA errors.

c Our estimates for the same 4,212 genes studied by Schwanhausser et al. (2011) derived using measured translation rates from Subtelny et al. (in press).

Direct measurements of translation rates support our analysis

Direct measurements of system wide translation rates using ribosome profiling (Guo et al., 2010; Ingolia et al., 2011; Subtelny et al., in press) provide independent evidence that translation rates vary less than Schwanhausser et al. (2011) suggest. The distributions of the rates of translation rates measured in mouse embryonic stem cells, mouse neutrophils, mouse NIH3T3 cells and human HeLa cells are all significantly narrower than Schwanhausser et al. (2011) inferred for mouse NIH3T3 cells (Fig. 5A; Table S1). For NIH3T3 cells the translation rates measured by ribosome profiling for 95% of the genes detected vary only 5.8 fold, but the rates inferred for 95% of genes by Schwanhausser et al. (2011) vary 115 fold (Fig. 5A). Because each of these datasets contain differing numbers of genes (Table S1), to provide a more direct comparison we took the intersection of genes detected by Schwanhausser et al. (2011) and by ribosome profiling in NIH3T3 cells (Fig. 5B). The variance in measured translation rates for the genes in the intersection is only 12% of the variance in rates inferred by Schwanhausser et al. (2011) for these same genes (Fig. 5B; Table S1).

Figure 5 Measured versus inferred translation rates.

(A) The relative density of ribosomes per mRNA for each gene directly measured by ribosome profiling (Guo et al., 2010; Ingolia et al., 2011; Subtelny et al., in press) (colored lines) compared to the translation rates for each gene inferred by Schwanhausser et al. (2011) (black lines). The distribution of values from the ribosome profiling experiments was scaled proportionally to have the same median as that of the Schwanhausser et al. (2011) values, and the gene frequencies of the each distribution was normalized to have the same total. The locations of the 2.5 and 97.5 percentiles of the two distributions for NIH3T3 cells are shown as dashed lines. (B) As panel A. except that the data for all genes in the Schwanhausser et al. (2011) dataset are shown in the solid black line and data for the genes in the intersection of the Schwanhausser et al. (2011) and Subtelny et al.’s (in press) datasets are shown in dashed lines. The variances and numbers of genes for each dataset are given in Table S1.

Having direct measurements of the variance in translation rates opens up a second strategy to estimate the relative importance of each step in gene expression (Materials and Methods). In our first strategy—the measured protein error strategy—protein degradation rates and errors in protein and mRNA abundances were determined from direct experimental data; and the variance in true protein levels explained by translation was inferred as that part of the variance in the residuals in Figure 4B that is not explained by the three experimentally measured terms. In our second strategy—the measured translation strategy—translation rates, protein degradation rates and mRNA errors are determined from direct experimental data; and the variance in measured protein levels explained by protein error is inferred as that part of the variance in the residuals in Figure 4A that is not explained by the sum of variances of the three experimentally measured components (Materials and Methods). This measured translation strategy is thus independent of our rescaling of Schwanhausser et al.’s (2011) second protein abundance estimates and of our estimate of stochastic protein measurement error.

According to our second strategy, for NIH3T3 cells the variance in true protein levels is 63% of the variance in Schwanhausser et al.’s (2011) measured protein abundances; mRNA levels contribute 81% to the variance in true protein expression; transcription 71%; RNA degradation 10%; translation 11%; and protein degradation 8% (Table 1). Despite the significant differences in the underlying data and assumption used, these results agree broadly with those of our first strategy (Table 1). Both strategies suggest that the variance in Schwanhausser et al.’s (2011) second protein abundance estimates is too high. Both suggest that translation contributes less to protein levels and that transcription contributes more that Schwanhausser et al. (2011) claimed. In effect, the measured rates of translation provide independent support for our rescaling of Schwanhausser et al.’s (2011) protein abundances and our estimates of stochastic protein error, and visa versa.

Our second strategy, though, does estimate that mRNA levels and transcription explain a higher percent of protein expression than the first (Table 1), but this is not entirely unexpected. In our first strategy, we were not able to take account of systematic, molecule specific errors in protein abundances because appropriate control measurements were not available. Thus, this first strategy could well have underestimated error. In contrast, our second strategy estimates all types of protein abundance errors in a single term and thus has the potential to be the more accurate if the error in the ribosome profiling and protein degradation data is not too large.

To further explore the relationship between our two strategies, we compared the correlation between translation rates inferred by Schwanhausser et al. (2011) and those measured by ribosome profiling in NIH3T3 cells (Fig. 6). The coefficient of determination is small (R2 = 0.13), indicating that the ribosome profiling data explain only 13% of the variance in Schwanhausser et al.’s (2011) inferred rates. Considered in isolation this result does not establish if the poor correlation is due to errors in either or both datasets. However, our measured protein error strategy shows that the variance in true translation rates contributes no more than 19% to the variance in Schwanhausser et al.’s (2011) inferred translation rates, with the remaining 81% of the variance being due to experimental error (Table 1; 0.19 = (0.34 × 0.30)∕(0.97 × 0.55)). The close agreement of this estimate with the actual correlation between measured and inferred translation rates (R2 ≤ 0.19 versus R2 = 0.13) suggests that the poor correlation is almost entirely due to error in Schwanhausser et al.’s (2011) inferred rates. In addition, this result provides further evidence that our two strategies broadly agree, with the measured protein error strategy potentially underestimating the degree of error in Schwanhausser et al.’s (2011) data.

Figure 6 Correlation between measured versus inferred translation rates.

The relationship between the measured rates of translation determined by Subtelny et al. (in press) using ribosome footprinting versus the inferred rates of translation determined by Schwanhausser et al. (2011) for the same set of 3,126 genes in NIH3T3 cells, see Table S1 for further details. The units shown are those provided in the original datasets. The linear regression is shown.

Ribosome profiling has also shown that translation rates change only several fold upon cellular differentiation and, with the exception of the translation machinery, the change affects all expressed genes to a similar degree (Ingolia et al., 2011). Other systemwide studies, including a separate analysis by Schwanhausser et al. (2011), also suggest that the differential regulation of translation may be limited to modest changes at a subset of genes (Baek et al., 2008; Hsieh et al., 2012; Kristensen et al., 2013; Schwanhausser et al., 2011; Selbach et al., 2008). This work seems consistent with our analysis and suggests that translation may be used chiefly to fine tune protein expression levels.

Estimating the number of non-transcribed genes

Both Schwanhausser et al.’s (2011) and all of our analyses presented above consider only those genes whose protein and mRNA expression was detected. There are many thousands of other genes, however, which express no mRNA and as a result cannot be translated. To estimate the proportion of such genes in a typical cell, we made use of a detailed analysis by Hebenstreit et al. (2011), Hebenstreit et al. (2012), who showed that there is a trimodal distribution of mRNA expression when the data is derived as an average for a population of cells of a single cell type (Figure S1). The first mode contains Highly Expressed (HE) genes, present at one or more molecules per cell; the second mode is comprised of Low Expressed (LE) genes, which are not expressed in most cells but—as shown by single molecule fluorescent in situ hybridization—are present at one to several molecules per cell in a small percent of cells; and the third mode contains genes that are not detectably expressed (NE genes) and thus, given the assays sensitivity, are present at less than one mRNA molecule per 100 cells. LE genes tend to be closer to HE genes on the chromosome than are NE genes, and it has been suggested that this proximity may allow escape from repressive chromatin structures in a few cells, explaining the stochastic bursts of rare transcription observed (Hebenstreit et al., 2012; Hebenstreit et al., 2011).

To account for variation in the expression of individual genes between cells, which all LE genes at a minimum must suffer, we assume that the general distribution of mRNA expression levels does not vary from cell to cell even when the expression of individual genes does. The mRNA expression of each LE gene was divided into a component representing expression of one mRNA molecule in some cells and a second component representing the remaining cells that express no mRNA (Materials and Methods). This yields 8,763 NE and LE gene equivalents that are not expressed and 12,546 LE and HE gene equivalents that are expressed. For the 8,763 non-expressed gene equivalents, the complete absence of their mRNAs from the cell means that they are not being translated in these cells. Therefore, there can be no variation in the rates at which they are translated. Instead, we assume that the absence of transcription is overwhelmingly the reason why these genes express no protein.

Implication for other system wide studies

Two other systemwide estimates of protein abundance in mammalian cells are, like Schwanhausser et al.’s (2011), lower than ours. These two reports suggest that the median abundance protein detected is present at 8,000 (Vogel et al., 2010) or 9,700 (Beck et al., 2011) molecules per cell versus our estimate of 170,000 molecules per cell. Since these lower estimates provide less than 1/10th of the number of histones needed to cover the diploid genome with nucleosomes and are lower than published estimates for a wide array of other housekeeping proteins, it is unlikely that they are accurate.

Another study by Wisniewski et al. (2012) provided protein abundance estimates for HeLa cells that are generally higher than ours and spread over a broader range (Fig. 7A). These estimates are 240% higher on average than the set of individual protein measurements from the literature (Dataset S3, Fig. 7B). Since over 80% of these individual measurements were made for proteins in HeLa cells, Wisniewski et al.’s (2012) estimates must be incorrectly scaled. Using our two part linear regression strategy, we therefore corrected Wisniewski et al.’s (2012) whole proteome data (Materials and Methods, Figure S2; Dataset S3), bringing the average variation between the whole proteome estimates and individual protein measurements to within 6% of each other (Fig. 7B; Dataset S3). Interestingly, the correction dramatically increases the similarity between the distributions of protein abundances in HeLa and NIH3T3 cells for all orthologous proteins (Fig. 7A). This establishes the important point, mentioned at the beginning of the Results: in aggregate the 60+ housekeeping proteins show a similar relationship to the expression values of all other cellular proteins in both cell lines, and thus the discrepancies with the uncorrected whole proteome data are not due to differences in expression levels in HeLa versus NIH3T3 cells. The correction also increases the correlation between HeLa cell protein and HeLa mRNA abundances to a statistically significant extent (p-value, 6 × 10−20) and reduces the 50% and 95% confidence bounds for this relationship by 1.7 fold and 4.6 fold respectively. Wisniewski et al. (2012) scaled their protein abundances using the total cellular protein content and the sum of the mass spectrometry signals for all detected polypeptides. They assumed that mass spectrometry signals are proportional to protein abundance. In contrast, our scaling strategy makes no such assumption and instead uses many individual measurements of housekeeping proteins to estimate a multipart (spline) function. The increased correlations obtained with individual protein measurements and with mRNA abundances for two cell lines suggests that our scalings are the more accurate.

Figure 7 Comparison of corrected and uncorrected whole proteome abundance estimates.

(A) The distributions of protein abundance estimates for 4,680 orthologous proteins in NIH3T3 cells (black lines) or HeLa cells (red lines). The values from Schwanhausser et al.’s (2011) second estimates and Wisniewski et al.’s (2012) estimates are shown as dashed lines. The values for our corrected abundance estimates are shown as solid lines. (B) The ratios of HeLa cell whole proteome abundance estimates divided by individual measurements from the literature for 66 proteins. Results for the original data from Wisniewski et al. (2012) (dashed line) and after these values have been corrected (solid line) are plotted. The green dashed vertical line indicates a ratio of 1.

Other estimates for the contribution of mRNA levels in determining protein expression in mammals are lower than ours, suggesting that mRNA levels contribute 10%–40% (Maier, Guell & Serrano, 2009; Vogel & Marcotte, 2012). In comparison, we estimate that mRNA abundance explains 56%–81% for a set of 4212 detected proteins. We also have suggested that for the 40% of genes in a given cell that express no mRNA, translation rates likely play no role in determining protein expression levels. The other groups neither took systematic experimental errors into account or made use of direct measures of translation rates and generally do not discuss non-transcribed genes. For this reason, their likely analyses underestimate the contribution of transcription.

Conclusions

Quantitative whole proteome analyses can offer profound insights into the control of gene expression and provide baseline parameters for much of systems biology. As these important new technologies continue to be refined, it is critical that the data be correctly scaled, that experimental errors be measured and accounted for as much as possible, that all genes be considered, and that direct measurements of each step in gene expression be made. Additional measurements and controls will be needed to derive a more assured systemwide understanding of protein and mRNA abundances and the relative importance of each of the four steps in gene expression.

Materials and methods

Correcting protein abundance

For NIH3T3 cells, all credible individual protein abundance measurements available to us for housekeeping proteins (a total of 61 proteins, Dataset S1) were log10 transformed along with the corresponding estimates from Schwanhausser et al.’s (2011) second whole proteome dataset. Model selection of different regressive models by leave-one-out cross-validation was used to fit the training data (Bickel & Doksum, 2001). This showed that a plausible two-part linear regression with a change point at 106 molecules per cell (line < 1 × 106… slope = 0.56, intercept = 2.64; line > 1 × 106… slope = 1.06, intercept = −0.41) fit the data far better than by chance (likelihood ratio test bootstrap p-value = 0.002 Bickel & Doksum, 2001; Figs. 3B and 3C). The resulting two-part linear model was used to correct all 5028 protein abundance estimates (Fig. 2B, Dataset S1).

The null hypothesis that the correlation coefficient of the uncorrected Schwanhausser et al. (2011) protein abundance estimates versus mRNA estimates (R1 = 0.626) is equal to that of our corrected protein estimates versus mRNA estimates (R2 = 0.642) was tested. The method for comparing dependent correlation coefficients (Olkin & Finn, 1990) was employed because both correlations involve the same mRNA-seq data and it is reasonable to assume that the uncorrected and corrected protein abundance estimates and the mRNA estimates have a multivariate Gaussian distribution. The resulting two-sided p-value < 10-29 shows that R2 is significantly larger than R1.

To correct protein abundance estimates for HeLa cells (Wisniewski et al., 2012), the same strategy used for NIH3T3 cells was used. A two-part linear regression with a change point at 106.8 molecules per cell fit the data far better than by accident (likelihood ratio test bootstrap p-value = 0.001) (Figure S2). The resulting two-part linear model was used to correct all HeLa cell protein abundance estimates (Fig. 7; Dataset S3). The correlation of HeLa cell protein abundance estimates with mRNA abundances was determined using the mean values of replica HeLa cell RNA-Seq datasets from the ENCODE consortium (The ENCODE Project Consortium, 2011) (GEO Accession ID GSM765402). The hypothesis that our corrected protein abundances correlate more highly with these HeLa mRNA abundances than the uncorrected estimates was tested as above, resulting in a two sided p-value of 6 × 10-20.

The contribution of mRNA to protein levels: measured protein error strategy

The variance term in a linear model between measured protein abundance (MP) (response) and measured mRNA levels (MR) (predictor) is decomposed in a standard way (ANOVA; Bickel & Doksum, 2001) into three components (Fig. 8). These components of the variance in the residuals represent mRNA measurement error (eR), protein measurement error (eP), and the variance in a linear model between true protein abundance (TP) and true mRNA levels (TR) that results from the centered genuine differences in the rates of protein degradation and translation (PDT). The measured protein abundances considered in this case are our rescaled estimates.

Figure 8 The relationship between true and measured protein and mRNA levels.

Statistically, we can write three linear models from Fig. 8 (1) TR=bRMR+cR+eR

(2) TP=bTR+c+PDT

(3) MP=TP+cP+eP

where TR, MR, TP, MP are abundance values on a log 10 scale; the three sources of variation (eR, eP and PDT) are assumed to be independent random variables with mean 0; the amount of protein degradation and translation (PDT) is taken to be independent of true mRNA levels (TR) on the basis of partial evidence: the variance in the residuals in Fig. 4B is similar for different mRNA abundances; the reversal of the causal relationship between TR and MR in model (1) assumes that TR and MR have an approximately joint Gaussian distribution; the slope of TP in model (3) is assumed to be 1 because the ratios between the 61 protein published abundance measurements and our corrected estimates are close to 1 (Fig. 2B); and finally we note that implicit in the analysis of variance is the assumption that the various datasets employed can be thought of as originating from a relatively homogeneous superpopulation. Combining (1)–(3), we write the linear model between measured protein abundance and measured mRNA levels as (4) MP=bbRMR+bcR+c+cP+beR+PDT+eP

Based on model (4)

i. We first estimated as var(beR + PDT + eP) as σall2 and bbR as b^all from fitting the above model with the 8,424 corrected mass spec and RNA-Seq data points pooled from the two replicates (Dataset S1). By independence, we have var(beR+PDT+eP)=b2var(eR)+var(PDT)+var(eP).

ii. We next estimated var(eR) as σ^R2 and bR as b^R from fitting model (1) with the 77 NanoString (‘TR’) versus RNA-Seq (‘MR’) data points, after removing two outliers (Dataset S2).

iii. We could not estimate var(eP) from directly fitting model (3), as TP data is not available. As a surrogate, we estimated var(eP) as σ^P2 from the following linear model that quantifies the stochastic error in mass spec replicate data: (5) MPij=avgMPi+(eP)ij,j=1,2

where MPij is the corrected mass spec data for the ith protein in the jth replicate in Schwanhausser et al. (2011), and avgMPi is the average of our corrected protein data for the ith protein, i = 1,…, 4212 (Dataset S1). Please note that σ^P2 is likely an underestimate of the protein error as we only consider the stochastic error, not the systematic error.

iv. From the estimates σall2, b^all, σ^R2, b^R and σ^P2 above, we estimate var(PDT) as σ^PDT2=σ^ all2−(b^allb^R)2σ^R2−σ^P2.

Hence, we have successfully decomposed the variance estimate, σ^all2 i.e., the estimated variance of residuals between measured protein levels and measured mRNA levels, into 3 components:

∙ σ^R2—RNA error (23.3% of σall2)

∙ σ^P2—protein error (7% of σall2)

∙ σ^PDT2—protein degradation and translation (69.6% of σall2).

From the diagram and the above calculation, we also derived the percentage of variability in the unobserved true protein levels explained by the unobserved true mRNA levels. σ^MP2−σ^P2−σ^PDT2σ^MP2−σ^P2=55.9%

where σ^MP2 is the variance of the corrected measured protein levels.

We separately estimated the stochastic mRNA error from the replicate RNA-Seq measurements of the 4,212 genes (Dataset S1). The stochastic mRNA error contributes 0.8% of σall2.

The contributions of transcription, translation and protein and mRNA degradation: measured error strategy

To determine the relative contributions of measured RNA degradation (RD) and measured protein degradation (PD) to the variance in true protein expression (TP), we estimated their variances, var(RD) and var(PD). We took Schwanhausser et al.’s (2011) calculated percentages for the contribution of RD and PD to explain the variance of their uncorrected mass whole proteome abundances (6.4% for RD and 4.9% PD, M Selbach, personal communication). Since the variance of the 8,424 uncorrected mass spec data points from the two replicates is 0.97, we thus calculated var(RD) and var(PD) as 0.062 and 0.048 respectively. The relative contributions of var(RD) and var(PD) to var(TP) (estimated as σ^MP2−σ^P2) was calculated (Table 1). We also determined the contribution of transcription (var(TXN)) to var(TP) as (var(TR) − var(true RD))∕var(TP), where var(TR) was estimated as σ^MP2−σ^P2−σ^PDT2, and the contribution of translation as (var(TP) − var(TR) − var(true PD))∕var(TP) (Table 1).

The contributions of each step of gene expression to protein levels: measured translation strategy

We calculated the relative contributions of each of the four steps in gene expression by an independent, second approach that does not rely either on our rescaling of Schwanhausser et al.’s (2011) protein abundance estimates or on our estimate of stochastic protein errors. Instead, our second approach infers true protein abundance based on Subtelny et al.’s (in press) direct measurements of translation rates in NIH3T3 cells by ribosome profiling (Subtelny et al., in press) and on our estimate of RNA measurement error. The measured protein abundances considered are thus Schwanhausser et al.’s (2011) second estimates, not our rescaling of these estimates. A central assumption is that since the variance in Subtelny et al.’s (in press) measured translation rates is 12% of the variance in the rates of translation inferred by Schwanhausser et al. (2011), then the contribution of translation to the variance in true protein levels is 12% of the value provided by Schwanhausser et al. (2011).

The variance term in a linear model between measured protein abundance (MP) and measured mRNA levels (MR) was decomposed as before (Fig. 8) except that the variance in the linear model between true protein abundance (TP) and true mRNA levels (TR) that results from the variance in the rates of protein degradation (PD) and protein translation (PT) were considered separately as cPD and dPT respectively. Similar to our measured error strategy, we can write three linear models using the same assumptions. (6) TR=bRMR+cR+eR

(7) TP=bTR+cPD+dPT+f

(8) MP=TP+cP+eP.

Thus, we can write the linear model between measured protein abundance (MP) and measured mRNA levels (MR) for the measured translation strategy as (9) MP=bbRMR+bcR+f+cP+beR+cPD+dPT+eP.

Based on this revised model (9).

i. We first estimated var(beR + cPD + cPT + eP) as σ^all2 and bbR as b^all from fitting the above model with the 8,424 mass spec and RNA-Seq data points pooled from the two replicates using Schwanhausser’s second estimates (Dataset S1). By independence, we thus have var(beR+cPD+cPT+eP)=b2var(eR)+var(cPD)+var(dPT)+var(eP).

ii. The values of var(eR) and bR are the same as those derived previously by our measured error strategy. Thus, we can estimate b^=b^all∕b^R.

iii. We used the estimate of var(cPD) from Schwanhausser et al. (2011), i.e., 0.97 × 5% = 0.0475.

iv. From Schwanhausser et al.’s (2011) results, we have var(dPT) = d2var(PT) estimated as 0.97 × 55% = 0.54. From Schwanhausser et al.’s (2011) estimates for each of 3,633 genes (Dataset S1, second tab, column AG) has an estimate of 0.29. Hence, the estimate of d2 is 1.86. From Subtelny et al. (in press), we have a separate, directly measured estimate of var(PT) as 0.03533, which we obtained by slightly increasing the variance of their data for the 3,126 genes in the intersected dataset (Fig. 5B; Table S1) by the ratio of the variances for Schwanhausser et al.’s (2011) inferred rates for the 3,633 genes and the 3,126 genes (Table S1). Using this value to replace that of Schwanhausser et al. (2011), we obtained a new estimate of var(dPT) = d2var(PT) as 1.86 × 0.03533 = 0.06593132.

v. Now we can estimate var(eP) as σ^P2=σ^all2−b^σ^R2−σ^cPD2−σ^dPT2 where σ^cPD2 is an estimate of var(cPD) and σ^dPT2 an estimate of var(dPT).

vi. Given Schwanhausser et al.’s (2011) second 8,424 uncorrected mass spec data, we can also estimate var(TP) as σ^TP2=σ^MP2−σ^P2, where σ^MP2 is an estimate of var(MP).

Given the estimates σ^cPD2 and σ^dPT2 and Schwanhausser et al.’s (2011) estimate of the contribution of the variance in RNA degradation (defined as σ^gRD2), we can decompose σ^TP2 as:

∙ variance explained by PD: σ^cPD2∕σ^TP2

∙ variance explained by PT: σ^dPT2∕σ^TP2

∙ variance explained by TR: 1−σ^cPD2σ^TP2−σ^dPT2σ^TP2

∙ variance explained by RD: σ^gRD2∕σ^TP2

∙ variance explained by TXN: 1−σ^cPD2σ^TP2−σ^dPT2σ^TP2−σ^gRD2σ^TP2.

The number of genes not transcribed in a typical cell within a population

To estimate the number of genes not transcribed in a typical cell within a population, we employed a deep RNA-Seq dataset that detected polyA + mRNA for 15,325 protein coding genes in mouse Th2 cells (Hebenstreit et al., 2011). To place these abundance estimates on the same scale as those of Schwanhausser et al.’s (2011) data the 3841 mRNAs expressed above 1 RPKM (reads per kilobase of exon per million mapped reads) in common between the two datasets were identified. The Th2 cell data were then scaled to have the same median and variance for these common genes in numbers of mRNA molecules per cell (Figure S3). Following Hebenstreit et al. (2012), we divided the expressed genes into 11,301 Highly Expressed (HE) genes, present at one or more mRNA molecule per cell, and 4,024 Low Expressed (LE) genes, expressed below one molecule per cell. The remaining 5,984 genes whose expression was not detected were designated Not Expressed (NE) genes. We then divided each LE gene into two: a fraction of a gene expressed at 1 molecule per cell with a weight w and a fraction of a gene that is not expressed in any cells with a weight 1 − w. The 4,024 LE genes were thus decomposed into 1,245 gene equivalents expressed at 1 molecules per cell and 2,779 gene equivalents that are not expressed. Combining these with the 11,301 HE genes and 5,984 NE genes, we obtained 12,546 HE and LE expressed gene equivalents and 8,763 NE and LE non-expressed gene equivalents.

Supplemental Information

Supplemental Information 1 Dataset S1

Click here for additional data file.

Supplemental Information 2 Dataset S2: data from Schwanhausser et al.’s nanostring and protein abundance calibration experiments

Click here for additional data file.

Supplemental Information 3 Dataset S3

Click here for additional data file.

Supplemental Information 4 The trimodal distribution of mRNA expression levels in animal cells.

The black curve shows the frequency distribution for 15,325 genes that give detectable polyA+ mRNA expression in mouse Th2 cells (Hebenstreit, Fang et al. 2011; Hebenstreit, Deonarine et al. 2012). The two major modes detected for these genes are Highly Expressed (HE) genes centered at 10 molecules of mRNA per cell and Low Expressed (LE) genes centered at 0.1 molecules per cell. The relative frequency of the remaining 5,984 Not Expressed (NE) genes is represented by the area of the circle. The grey curve shows the expression frequency distribution in Th2 cells of the 3,841 genes expessed above 1 molecule per cell that are from the set of the 4,212 genes whose mRNA and protein abundances were detected by Schwanhausser et al. All data has been scaled as described in the Materials and Methods and Figure S3.

Click here for additional data file.

Supplemental Information 5 Calibrating absolute protein abundances in HeLa cells.

a, The relationship between individually derived estimates for 66 housekeeping proteins (y axis) and Wisniewski et al.s whole proteome estimates from HeLa cells (x axis) (Dataset S3). The two part line of best fit used to correct the whole proteome estimates is shown (solid red line) as is the single linear regression (dashed red line). b, The fit of different regression models for the data in panel a. The y axis shows the leave-one-out cross validation root mean square error for each model. The x axis shows the protein abundance used to separate the data for two part linear regressions. The red curve shows the optimum change point for a two part linear model is at an abundance of 106.8 molecules per cell. The dashed red horizontal line shows the root mean square error for the single linear regression.

Click here for additional data file.

Supplemental Information 6 Scaling Hebenstreit et al.s mRNA abundances.

The distribution of mRNA abundnaces from three datasets are shown. The 3,841 mRNAs expressed above 1 RPKM in the Hebenstreit et al. RNA-Seq data that are in common with mRNAs detected by Schwanhausser et al were identified (dashed red line). These abundances were then scaled to have the same median and variance as Schwanhausser et al.s data (solid red line). This scaling was in addition applied to all other genes in the Hebenstreit et al. data and the resulting values used in the simulation shown in Figure 6 and in the mRNA expression distribution shown in Figure S1.

Click here for additional data file.

Supplemental Information 7 Variances in translation rates estimates

Click here for additional data file.

We are indebted to Matthias Selbach for providing his second whole proteome abundance estimates and ancillary data from the Schwanhausser et al. (2011) analysis. We acknowledge his patient answering of our questions about the Schwanhausser et al. (2011) paper. We are particularly grateful to Stephen Eichhorn and David Bartel for generously providing their ribosome profiling data for NIH3T3 cells prior to publication. We also thank Sarah Teichmann for helping us better understand the Hebenstreit et al. (2012) analysis of mRNA expression and Susan Celniker, Ben Brown, and David Knowles for constructive comments on our manuscript.

Additional Information and Declarations

Competing Interests

Author Contributions

Mark Biggin is an Academic Editor for PeerJ. The authors declare that they have no competing interests.

Mark D Biggin conceived and designed the experiments, performed the experiments, wrote the paper.

Jingyi Jessica Li conceived and designed the experiments, performed the experiments, analyzed the data, contributed analysis tools, wrote the paper.

Peter J Bickel conceived and designed the experiments, wrote the paper.

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
