# Peer review of "System wide analyses have underestimated protein abundances and the importance of transcription in mammals"

_PeerJ, doi:10.7717/peerj.270_

## Round 0.1 · original submission · Major Revisions

The reviewer did a very careful job of reading and evaluating the manuscript. The major recommended changes are straightforward, in particular how results are reported for non-expressed genes, and clarifying the difficulties associated with specific experimental techniques.

Reviewer 1 ·

Basic reporting

See below.

Experimental design

See below.

Validity of the findings

See below.

Additional comments

The authors present an argument that that the work of Schwanhausser et
al as well as other, previous analysis of absolute protein levels across genes have under estimated the role of transcription in determining protein levels. Although the specific merits of the quantitative analyses they preform could be debated, I think this is nice work that presents a controversial argument that should be published. However, there are major issues that the authors should address:

Major issues:

Issue 1. The title: "System Wide Analyses have Underestimated Protein
Abundances and Transcriptional Importance in Mammals"

The title "Transcriptional Importance" makes little sense. I would suggest the following title: "System Wide Analyses of Protein Abundance Across Genes have
Underestimated the Importance of Transcription"

Issue 2. The authors should remove all of the results with non-expresed genes. These inflate the authors estimates of the contribution of translation. For instance, R2 = 1 for non-expressed genes is meaningless. One could just say that all genes are not expressed as protein and this same result would be obtained. An ROC analysis would be better, but the proteome data only goes to a limited depth into the expressed proteome. The R2 of 0.96 reported in Figure 6 for all genes is driven by the nonexpressed genes. The protein data only allows sampling of up to a limited minimum copy number per cell.

Issue 3. One possible alternative analysis would be to consider a new data set of absolute protein abundances that goes deeper into the proteome such as:
Nagara et al. Deep proteome and transcriptome mapping of a human
cancer cell line Molecular Systems Biology 7 Article number: 548
doi:10.1038/msb.2011.81
and
Beck et al The quantitative proteome of a human cell line. Molecular
Systems Biology 7; Article number 549; doi:10.1038/msb.2011.82

Can the authors show the nonlinear effect in these data as well?

Issue 4. Can the authors answer: are the translation rates estimated by Schwanhausser et al correlated with the translation estimates obtained from ribosomal foot printing? Is the difference in variance driven by outliers?

Issue 5. The authors should describe the limitations of iBAQ and related measures of absolute namely that might not be very reproducible and that the absolute abundance of estimates of individuals genes cannot be trusted without additional verification. e.g. cite Peng et al. Protease bias in absolute protein quantitation. Nature Methods 9, 524–525 (2012) doi:10.1038/nmeth.2031
* * *
Minor comments:

How are the mRNA molecules per cell values derived? Are these just RPKM?

Figure 3 is mentioned before figure 2. "fit the data far better than
by accident" -> "fit the data far better than by chance"

Figure 3's caption should refer to Figure 4 regarding the 2-part correction.

I am unfamiliar with comparing R2 estimates. Can the authors describe
the test in more detail? The phrase does not make sense, but I think
I understand what the authors mean: "The resulting two-sided p-value <
10-29 shows that R2 is statistically significantly larger than R1."

please check these and similar phrases "statistically highly significant" -> "highly significant"

---

## Round 0.2 · Minor Revisions

I agree with the reviewer that the perspective of the article is important, and some overview of what the debate is about the current analyses, and also the limitations of their analysis, is essential. It also seems a trivial but necessary change to remove unexpressed genes, is this not so? When redoing analyses properly that others have undertaken, it is critical to have full transparency in this analysis so that the field may follow suit.

Reviewer 1 ·

Basic reporting

No comments.

Experimental design

See below.

Validity of the findings

See below.

Additional comments

I do not agree with the author's statement that "We would characterize our work as the unremarkable practice of ensuring that recent analyses are consistent with decades of well established data in the literature and applying commonly accepted statistical methods to estimate the impact of error."

The key point of debate is how these errors terms are estimated. Readers may not agree with the author's approaches for estimating these terms. The authors do not do a good job of explaining this to a broad audience, nor do they highlight the limitations of their analysis.

I reluctantly recommend publication of this work if the authors address the following comments:

Major comment: The authors should remove the analysis of non-expressed genes using data from Hebenstreit et al. It is unclear, unless they are being used to inflate their estimates, why this is even necessary. In particular, the last sentence in the abstract, "Since these genes cannot be translated, we argue that translation plays no role in determining their levels of expression," makes absolutely no sense. Why argue this? Why is it even necessary to include this result in this revision?

Additional comment 1: Mid-review the authors revised their estimates for the contribution of variation in mRNA levels to variation in protein levels from 75% to 81%. It is unclear how this new number was obtained and leaves me suspect of their analysis. The authors should explain this in more detail.

Additional comment 2: I would be more convinced of their results of they could use data that achieved higher coverage of the proteome e.g. Nagara et al. perform error modeling and obtain the same estimates of the contribution of variation transcription to variation in protein levels.

---

## Round 0.3 · accepted · Accept

I appreciate your thoughtful response to the reviewer and your clarification of these points in the manuscript. I am glad this analysis will be published.